# *yama*, a mutant allele of *Mov10l1*, disrupts retrotransposon silencing and piRNA biogenesis

Yongjuan Guan[1], Scott Keeney[2,3], Devanshi Jain[2,4]*, P. Jeremy Wang[1]*

**1** Department of Biomedical Sciences, University of Pennsylvania School of Veterinary Medicine, Philadelphia, Pennsylvania, United States of America, **2** Molecular Biology Program, Memorial Sloan Kettering Cancer Center, New York, United States of America, **3** Howard Hughes Medical Institute, Memorial Sloan Kettering Cancer Center, New York, United States of America, **4** Department of Genetics, Rutgers University, Piscataway, New Jersey, United States of America

* devanshi.jain@rutgers.edu (DJ); pwang@vet.upenn.edu (PJW)

**Data Availability Statement:** All relevant data are within the manuscript and its Supporting Information files.

**Funding:** Work in the Wang lab was supported by NIH/NICHD grants R01 HD069592 and P50

## Abstract

Piwi-interacting RNAs (piRNAs) play critical roles in protecting germline genome integrity and promoting normal spermiogenic differentiation. In mammals, there are two populations of piRNAs: pre-pachytene and pachytene. Transposon-rich pre-pachytene piRNAs are expressed in fetal and perinatal germ cells and are required for retrotransposon silencing, whereas transposon-poor pachytene piRNAs are expressed in spermatocytes and round spermatids and regulate mRNA transcript levels. MOV10L1, a germ cell-specific RNA helicase, is essential for the production of both populations of piRNAs. Although the requirement of the RNA helicase domain located in the MOV10L1 C-terminal region for piRNA biogenesis is well known, its large N-terminal region remains mysterious. Here we report a novel *Mov10l1* mutation, named *yama*, in the *Mov10l1* N-terminal region. The *yama* mutation results in a single amino acid substitution V229E. The *yama* mutation causes meiotic arrest, de-repression of transposable elements, and male sterility because of defects in pre-pachytene piRNA biogenesis. Moreover, restricting the *Mov10l1* mutation effects to later stages in germ cell development by combining with a postnatal conditional deletion of a complementing wild-type allele causes absence of pachytene piRNAs, accumulation of piRNA precursors, polar conglomeration of piRNA pathway proteins in spermatocytes, and spermiogenic arrest. Mechanistically, the V229E substitution in MOV10L1 reduces its interaction with PLD6, an endonuclease that generates the 5′ ends of piRNA intermediates. Our results uncover an important role for the MOV10L1-PLD6 interaction in piRNA biogenesis throughout male germ cell development.

## Author summary

Small non-coding RNAs play critical roles in silencing of exogenous viruses, endogenous retroviruses, and transposable elements, and also play multifaceted roles in controlling gene expression. Piwi-interacting RNAs (piRNAs) are found in gonads in diverse species

HD068157 (PJW). Work in the Keeney lab was supported by the Howard Hughes Medical Institute (SK). DJ was supported in part by a fellowship from the Human Frontier Science Program. Core facilities at MSK are supported by NCI Cancer Center Support Grant P30 CA008748. The funders had no role in study design, data collection and analysis, decision to publish, or preparation of the manuscript.

**Competing interests:** The authors have declared that no competing interests exist.

from flies to humans. An evolutionarily conserved function of piRNAs is to silence transposable elements through an adaptive mechanism and thus to protect germline genome integrity. In mammals, piRNAs also provide a poorly understood function to regulate postmeiotic differentiation of spermatids. More than two dozen proteins are involved in the piRNA pathway. MOV10L1, a germ-cell-specific RNA helicase, binds to piRNA precursors to initiate piRNA biogenesis. Here we have identified a single amino acid substitution (V229E) in MOV10L1 in the *yama* mouse mutant. When constitutively expressed as the only source of MOV10L1 throughout germ cell development, the *yama* mutation abolishes piRNA biogenesis, de-silences transposable elements, and causes meiotic arrest. When the mutant phenotype is instead revealed only later in germ cell development by conditionally inactivating a wild-type copy of the gene, the point mutant abolishes formation of later classes of piRNAs and again disrupts germ cell development. Point mutations in MOV10L1 may thus contribute to male infertility in humans.

## Introduction

Transposable elements, which constitute around 40% of the mammalian genome, play important roles in genome evolution. However, their mobilization can disrupt gene function and cause diseases [1,2]. Production of piRNAs in the germline is one of the major mechanisms to silence retrotransposons to protect genome integrity. piRNAs are small (26–31 nt) non-coding RNAs with a preference for a 5′ uridine nucleotide [3]. piRNAs associate with homologs of the *Drosophila melanogaster* Piwi protein, which in mouse include MIWI (PIWIL1), MILI (PIWIL2), and MIWI2 (PIWIL4). In the mouse germline, two populations of piRNAs are present: pre-pachytene and pachytene. Pre-pachytene piRNAs associate with MILI and MIWI2 in embryonic and perinatal germ cells and are required for DNA methylation and retrotransposon silencing [4–7]. Pachytene piRNAs are present in spermatocytes and round spermatids and are associated with MILI and MIWI [8–10]. In contrast with the predominant role of pre-pachytene piRNAs in silencing transposable elements, pachytene piRNAs are implicated in cleavage of messenger RNAs in testis [11–14]. The mRNA cleavage is dependent on the slicer activity of MIWI [14,15]. In addition, MIWI binds to and stabilizes spermiogenic mRNAs directly [16]. MIWI and pachytene piRNAs also function in activating translation of a subset of spermiogenic mRNAs [17]. Thus, piRNAs perform diverse functions throughout male germ cell development in mammals.

In addition to the Piwi proteins, more than 20 other proteins are involved in the piRNA pathway [3]. A number of Tudor domain-containing (TDRD) proteins (TDRD1 [18–20], TDRD5 [21,22], TDRD9 [23,24], TDRD12 [25], and TDRKH [26,27]), in general, function as scaffold proteins in the assembly of piRNA ribonucleoprotein particles by binding to arginine-methylated Piwi proteins [28–31]. The *Drosophila* Zucchini protein functions as an endonuclease in piRNA biogenesis [32–34], and its mammalian orthologue PLD6 is essential for piRNA biogenesis in mouse [35,36]. Zucchini/PLD6 cleaves precursor transcripts into intermediate piRNA fragments, whose 5′ ends are bound and protected by Piwi proteins; subsequently the Piwi-bound fragments are trimmed to the length of mature piRNAs by the 3′-to-5′ exoribonuclease PNLDC1 [37–40].

MOV10L1, a germ cell-specific RNA helicase, is a key regulator of piRNA biogenesis. MOV10L1 interacts with all Piwi proteins and binds to piRNA precursors to initiate primary piRNA biogenesis [41,42]. MOV10L1 helicase resolves G-quadruplex RNA secondary structures [42,43]. Constitutive inactivation of MOV10L1 by deletion of the helicase domain leads

to loss of mature piRNAs, de-repression of retrotransposons, arrest during meiotic prophase, and male infertility [41,44]. In contrast, conditional deletion of the helicase domain-encoding region of *Mov10l1* postnatally leads to arrest during postmeiotic spermatid differentiation, without overt defects in transposon silencing [45]. Two point mutations previously generated in the MOV10L1 C-terminal RNA helicase domain revealed an essential role for its RNA helicase activity in piRNA biogenesis [42,46]. However, the function of the large N-terminal region of MOV10L1 remains unknown, because it does not share any similarity with known protein domains. We address this question here using a *Mov10l1* V229E missense mutant recovered as part of a recently described *N-ethyl-N*-nitrosourea (ENU) mutagenesis screen for novel meiotic mutants [47,48]. This substitution attenuates the interaction between MOV10L1 and PLD6 and causes a profound failure in both pre-pachytene and pachytene piRNA biogenesis. We named this mutant allele *yama*, for '**y**ields **a**bnormal meiosis and **M**ov10l1-**a**ffected'. Other hits we previously described from the screen are *rahu, ketu and shani* [47–49]. Yama, Rahu, Ketu and Shani are harbingers of misfortune in Vedic mythology.

## Results

### *Mov10l1*<sup>yama/yama</sup> males exhibit meiotic arrest and sterility

The *Mov10l1*<sup>yama</sup> allele was isolated in an ENU mutagenesis screen for mutants with autosomal recessive defects in meiosis. Details of the screen are provided elsewhere [47,48]. Briefly, mutagenesis was performed on male mice of the C57BL/6J strain, and then a three-generation breeding scheme was carried out including outcrossing with females of the FVB strain (Fig 1A). Third-generation male offspring were screened for meiotic defects by immunostaining spermatocyte squash preparations for well-established meiotic markers SYCP3 and γH2AX, which enabled rapid assessment of defects in both synapsis and meiotic recombination.

In a screen line we named *yama*, nine of 27 third-generation males screened displayed abnormal SYCP3 and γH2AX immunostaining patterns indicative of meiotic defects (Fig 1B). The phenotype was first mapped using SNP genotyping arrays and manual genotyping of strain polymorphisms to a 17.44-Mb interval on Chromosome 15 (Fig 1C) [47]. Whole-exome sequencing was performed on DNA from one mutant, and un-annotated sequence variants within the mapped region were identified as previously described [47]. This analysis revealed a single exonic mutation located in *Mov10l1*. This variant is a T to A transversion at position Chr15:88994968 (GRCm38/mm10), resulting in a missense mutation in exon 5: V229E (codon GTG229GAG) (S1 Fig). Valine 229 in MOV10L1 is highly evolutionarily conserved (Fig 2A). As detailed below, the targeted *Mov10l1* mutation and the *yama* mutation confer similar phenotypes and fail to complement one another. We conclude that the ENU-induced V229E mutation disrupts MOV10L1 function and is the cause of the *yama* mutant phenotype.

*Mov10l1*<sup>yama/yama</sup> mice were viable and exhibited no obvious gross abnormalities. *Mov10l1*<sup>yama/yama</sup> females were fertile (7.3 ± 1.0 pups/litter, n = 4) but males were sterile. The body weight was comparable between 8-week-old *Mov10l1*<sup>yama/yama</sup> mice (24.2 ± 1.1 g, n = 3) and *Mov10l1*<sup>+/yama</sup> littermates (23.8 ± 0.9 g, n = 3). However, the testis weight of adult *Mov10l1*<sup>yama/yama</sup> males was sharply reduced in comparison with their heterozygous littermates (Fig 2B and 2C). We next examined histology of testes from juvenile and adult males at ages when testes are enriched for specific germ cell types. Germ cells enter meiosis in a semi-synchronous wave during the second week after birth [50]. At postnatal day 21 (P21), testes contain meiotic cells and round spermatids; elongated spermatids are present by P28; by adulthood (P60), several waves of meiosis have occurred and testes contain a mixed population of germ cells at all stages. *Mov10l1*<sup>+/yama</sup> P21 and P28 testes contained meiotic cells and spermatids, as expected, and adult (P60) testes contained a full spectrum of germ cells: spermatogonia, spermatocytes,

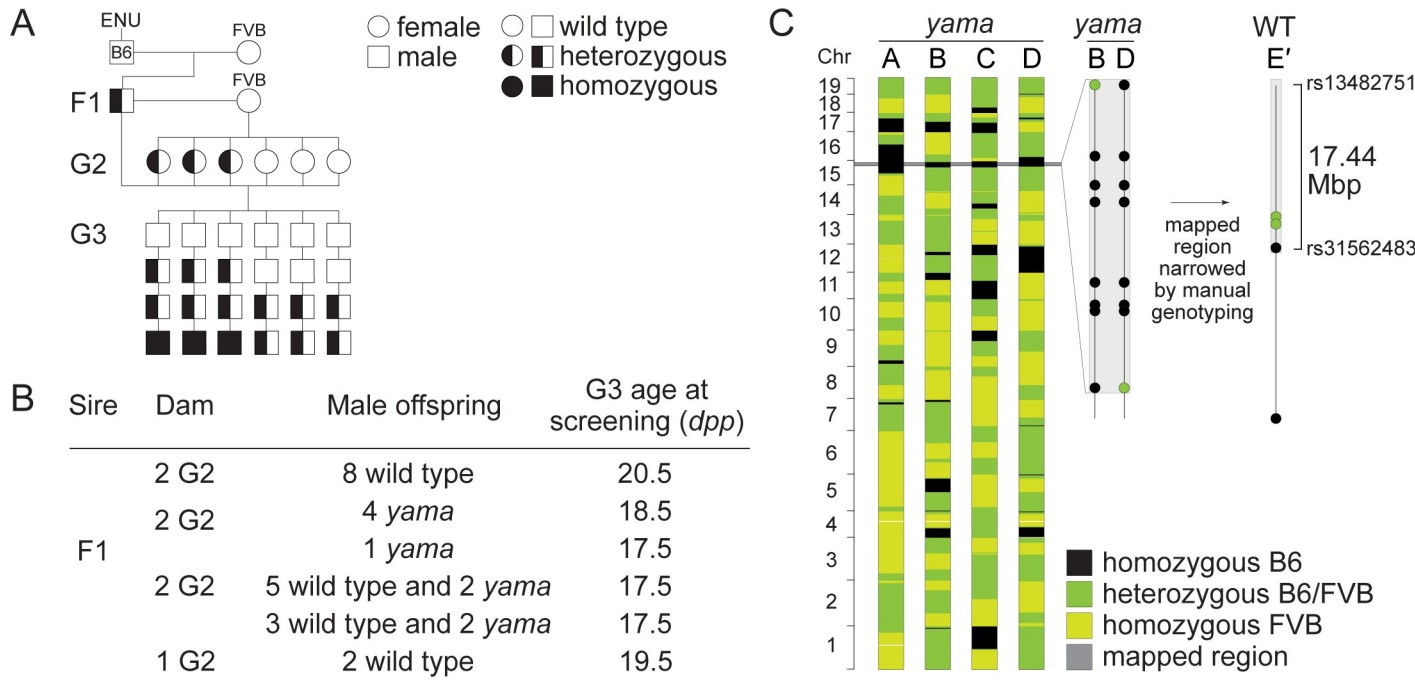

**Fig 1. Isolation of the *Mov10l1^yama^* allele.** (A) Breeding scheme used to isolate mutants with autosomal recessive defects in meiosis. Male mice of the C57BL/6J strain (B6) were mutagenized with ENU and bred to wild-type females of the FVB/NJ strain (FVB) to generate founder (F1) males that were potential mutation carriers. Each F1 male was bred to wild-type FVB females to produce second generation (G2) offspring, and then G2 daughters were crossed back to their F1 sire to produce third-generation (G3) offspring. G3 males were screened for meiotic defects. For a line carrying a single autosomal recessive mutation of interest, roughly one half of G2 females were expected to be carriers and one eighth of all G3 males were expected to be homozygous. (B) Screen results obtained for the *yama* line where the F1 male was harem-bred to seven G2 females. This generated 27 G3 males; 18 males were wild-type and nine males displayed the *yama* phenotype. (C) SNP genotypes of four G3 *yama* mutants (A, B, C, D) obtained using the Illumina Medium Density Linkage Panel are shown on the left. As mutagenesis was performed on B6 males, we expected ENU-induced mutations to be linked to B6 variants for strain polymorphisms that differed between B6 and FVB. We also expected G3 mutants to be homozygous for those same linked B6 variants. Therefore, we mapped the phenotype-causing mutation by searching for regions of B6 homozygosity that are shared between mutants. The single 32.45-Mbp region of B6 SNP homozygosity that is shared between mutants A, B, C, and D is highlighted in grey. A detailed view of variants within this region is shown in the middle for two informative *yama* mutants (B, D). We narrowed this region by manually genotyping more G3 mutants and phenotypically normal siblings for additional SNPs within this region. We expected phenotypically normal G3 mice not to be homozygous for the phenotype-causing mutation or its linked B6 variants. A detailed view of variants used to refine the mapped region is shown on the right for one informative phenotypically normal *mouse* (E′). The final mapped region was a 17.44-Mbp interval on Chromosome 15, flanked by heterozygous SNPs rs31562483 (Chr15:85684090) and rs13482751 (Chr15:102951530).

round spermatids, and elongated spermatids (Fig 2D). In contrast, *Mov10l1^yama/yama^* testes from P21, P28, and P60 all lacked post-meiotic spermatids (Fig 2D). While Sertoli-cell-only tubules were not observed in P60 or younger *Mov10l1^yama/yama^* testes, tubules with a Sertoli-cell-only phenotype or a severe loss of germ cells accounted for 25% of all tubules in 4-month-old (n = 2 mice) and 71% of all tubules in 7-month-old (n = 2 mice) *Mov10l1^yama/yama^* testes, showing a progressive loss of germ cells with age (S2 Fig).

Spermatocytes from mice homozygous for a targeted deletion of the helicase domain of *Mov10l1* (*Mov10l1^-/-^*) arrest early in meiosis leading to a complete lack of post-meiotic germ cells, hypogonadism and infertility [41,44]. Our initial observations indicated similar phenotypes for *Mov10l1^yama/yama^*, therefore we evaluated meiotic events more closely. We examined chromosomal synapsis in spermatocytes by immunofluorescence of nuclear spreads with antibodies against SYCP1 and SYCP3, components of the synaptonemal complex (SC). The SC is a tripartite proteinaceous structure that physically links homologous chromosomes during meiosis. SYCP3 is a component of the SC axial/lateral elements formed along the axis of each chromosome, and SYCP1 is a SC transverse element that connects the two lateral elements to synapse homologous chromosomes [51]. In control (*Mov10l1^+/yama^*) pachytene spermatocytes,

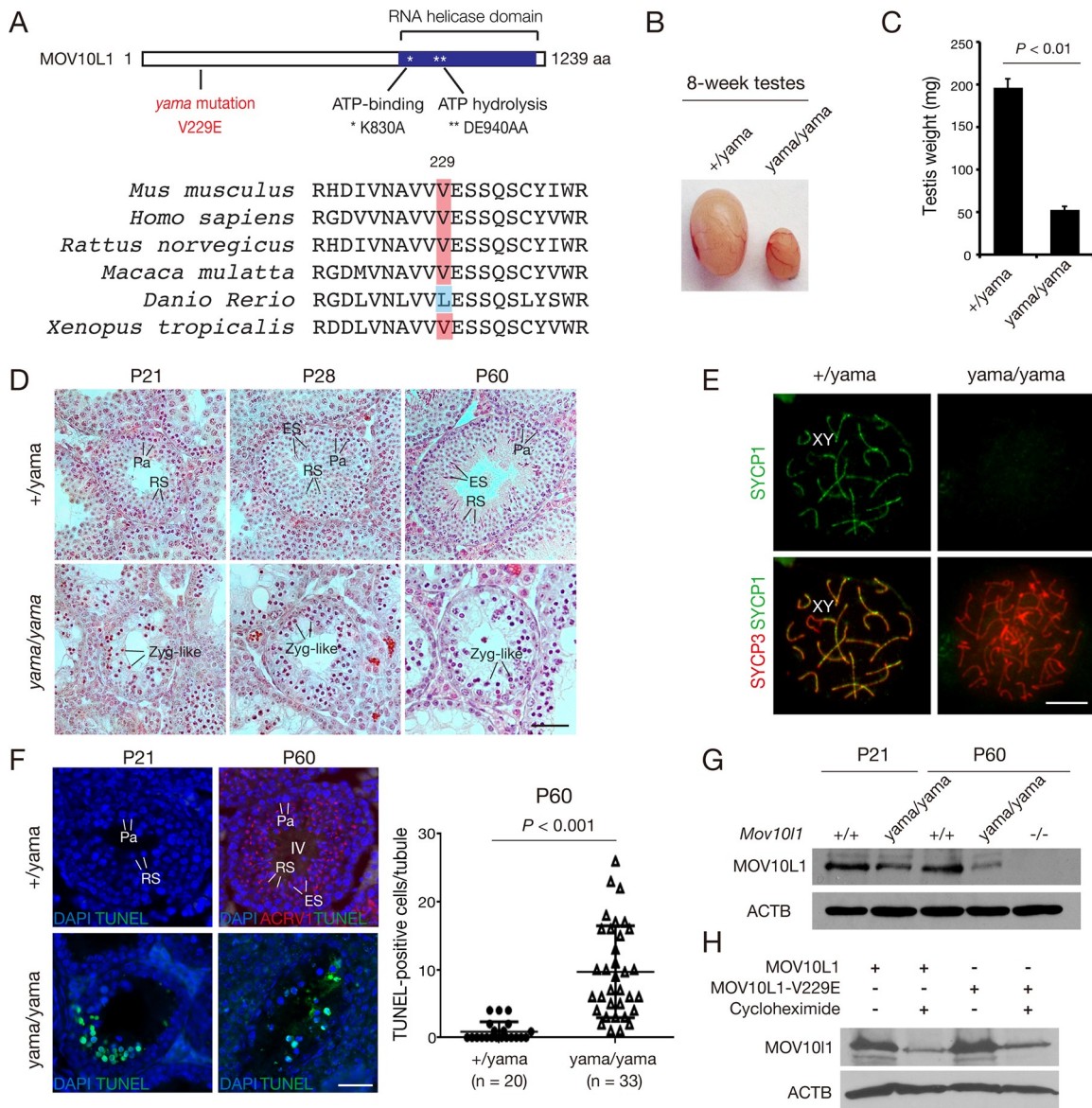

**Fig 2. The *Mov10l1* V229E mutation causes meiotic arrest and male sterility.** (A) Schematic diagram of the full-length mouse MOV10L1 protein (XP_006521619) and the conservation of the V229 residue. K830A and DE940.941AA mutations in the RNA helicase domain were previously reported and included for comparison [42,46]. (B) Dramatic size reduction of testis from 8-week-old *Mov10l1*[yama/yama] mice. (C) Reduction of testis weight in 8-week-old *Mov10l1*[yama/yama] mice (mean ± s.d.; n = 3 per genotype). (D) Histology (hematoxylin/eosin staining) of postnatal day 21 (P21), P28 and P60 testes from control and *Mov10l1*[yama/yama] mice. Abbreviations: Zyg-like, zygotene-like spermatocytes; Pa, pachytene spermatocytes; RS, round spermatid; ES, elongating spermatid. Scale bar, 50 μm. (E) Synapsis analysis of spermatocytes from P60 (adult) testes. Scale bar, 10 μm. (F) TUNEL analysis of seminiferous tubules from P21 and P60 testes. Quantification of TUNEL-positive spermatocytes at P60 is plotted. n, the number of stage IV tubules from *Mov10l1*[+/yama] testes or TUNEL-positive tubules from *Mov10l1*[yama/yama] testes. ACRV1, a marker of acrosome [67], is used for staging of seminiferous tubules in wild-type P60 testes. Scale bar, 50 μm. (G) Western blot analysis of MOV10L1 in P21 and P60 testes. *Mov10l1*[-/-] (knockout) testis serves as a negative control. ACTB serves as a loading control. (H) Western blot analysis of wild-type MOV10L1 and MOV10L1[V229E] in HEK293T cells. Transfected cells were collected after 24 hours with or without cycloheximide. Cycloheximide inhibits *de novo* protein synthesis so that protein stability can be compared in the absence of translation. The MOV10L1 antibody (G and H) was raised against the N-terminal region (amino acid 1–101) as previously reported [41].

all chromosomes were fully synapsed except the sex chromosomes (Fig 2E). In contrast, the most advanced *Mov10l1*<sup>yama/yama</sup> spermatocytes assembled the SC axial elements containing SYCP3 but lacked chromosomal synapsis as shown by the absence of SYCP1, and thus were considered zygotene-like spermatocytes (Fig 2E). This is consistent with our histological analyses where the most advanced spermatocytes in *Mov10l1*<sup>yama/yama</sup> appeared zygotene-like (Fig 2D). We interpret the absence of more advanced stages to be due to elimination of aberrant cells by apoptosis. Indeed, TUNEL analysis revealed that apoptosis was strongly increased in P21 (juvenile; when wild-type testes are enriched for meiotic cells and round spermatids) and in P60 (adult; when wild-type testes contain a full spectrum of germ cell types) *Mov10l1*<sup>yama/yama</sup> testes, suggesting that defective spermatocytes were eliminated in the mutants (Fig 2F). We conclude that the V229E mutation causes a blockade in early meiosis similar to that previously reported for the *Mov10l1*<sup>-/-</sup> mutant [41,44].

Western blot analysis showed that the abundance of MOV10L1 protein was modestly lower in *Mov10l1*<sup>yama/yama</sup> testis than wild-type at P21, and was more substantially reduced at P60 (Fig 2G). Because MOV10L1 is abundantly expressed in pachytene spermatocytes [41], and P21/P60 wild-type testes were rich in pachytene spermatocytes but the mutant testes were depleted of normal pachytene spermatocytes, the reduction of MOV10L1 abundance in the mutant testis could be due to depletion of spermatocytes. It is also possible that the MOV10L1 (V229E) protein was less stable. To test the latter possibility, we expressed MOV10L1 and MOV10L1$^{V229E}$ in HEK293T cells separately. The protein abundance of MOV10L1 and MOV10L1$^{V229E}$ was similar, indicating that the V229E substitution does not affect the intrinsic stability of MOV10L1 (Fig 2H).

## De-repression of LINE1 and IAP retrotransposons in *Mov10l1*<sup>yama/yama</sup> testis

Because MOV10L1-dependent biogenesis of pre-pachytene pRNAs is required for silencing of retrotransposons [41,42], we sought to address whether transposable elements were affected in *Mov10l1*<sup>yama/yama</sup> testes. We examined the expression of LINE1 and IAP in P21 and P60 *Mov10l1*<sup>+/yama</sup> and *Mov10l1*<sup>yama/yama</sup> testis sections by immunostaining with anti-LINE1 ORF1 and anti-IAP GAG antibodies. LINE1 and IAP were barely detected in the *Mov10l1*<sup>+/yama</sup> testes but were highly upregulated in the *Mov10l1*<sup>yama/yama</sup> testes (Fig 3A and 3B). Consistent with the upregulation patterns in *Mov10l1*-null testes [41], LINE1 was mainly upregulated in spermatocytes, whereas IAP was de-repressed in spermatogonia in *Mov10l1*<sup>yama/yama</sup> testes (Fig 3A and 3B). These results were confirmed by Western blot and quantitative RT-PCR analyses in *Mov10l1*<sup>+/yama</sup> and *Mov10l1*<sup>yama/yama</sup> testes at different ages: P10, P21, and P60 (Fig 3C and 3D). P10 testes are expected to contain spermatogonia and pre-leptotene spermatocytes but no later germ cell types. These data demonstrate that retrotransposons are de-repressed in *Mov10l1*<sup>yama/yama</sup> testes, similar to *Mov10l1*<sup>-/-</sup> mice. We infer that this is likely due to disruption of pre-pachytene piRNA biogenesis. The differential de-repression of LINE1 and IAP in postnatal testes also occurs in *Mili* or *Tex15*-deficient mice, suggesting the presence of additional retrotransposon-specific regulating mechanisms in different germ cells [52–54].

We next examined the expression and localization of MILI and MIWI, two postnatal Piwi proteins, in P21 and P60 *Mov10l1*<sup>+/yama</sup> and *Mov10l1*<sup>yama/yama</sup> testes. As expected, MILI was detected in both spermatogonia and pachytene spermatocytes in *Mov10l1*<sup>+/yama</sup> testis (Fig 3E). However, MILI was only detected in spermatogonia in *Mov10l1*<sup>yama/yama</sup> testes, which may reflect the absence of normal pachytene spermatocytes in the mutant testes (Fig 3E). MIWI was expressed in pachytene spermatocytes but not in spermatogonia in *Mov10l1*<sup>+/yama</sup> testis, and not detected in the remaining spermatocytes in *Mov10l1*<sup>yama/yama</sup> testes, which again may reflect the absence of normal pachytene spermatocytes in mutants (Fig 3F).

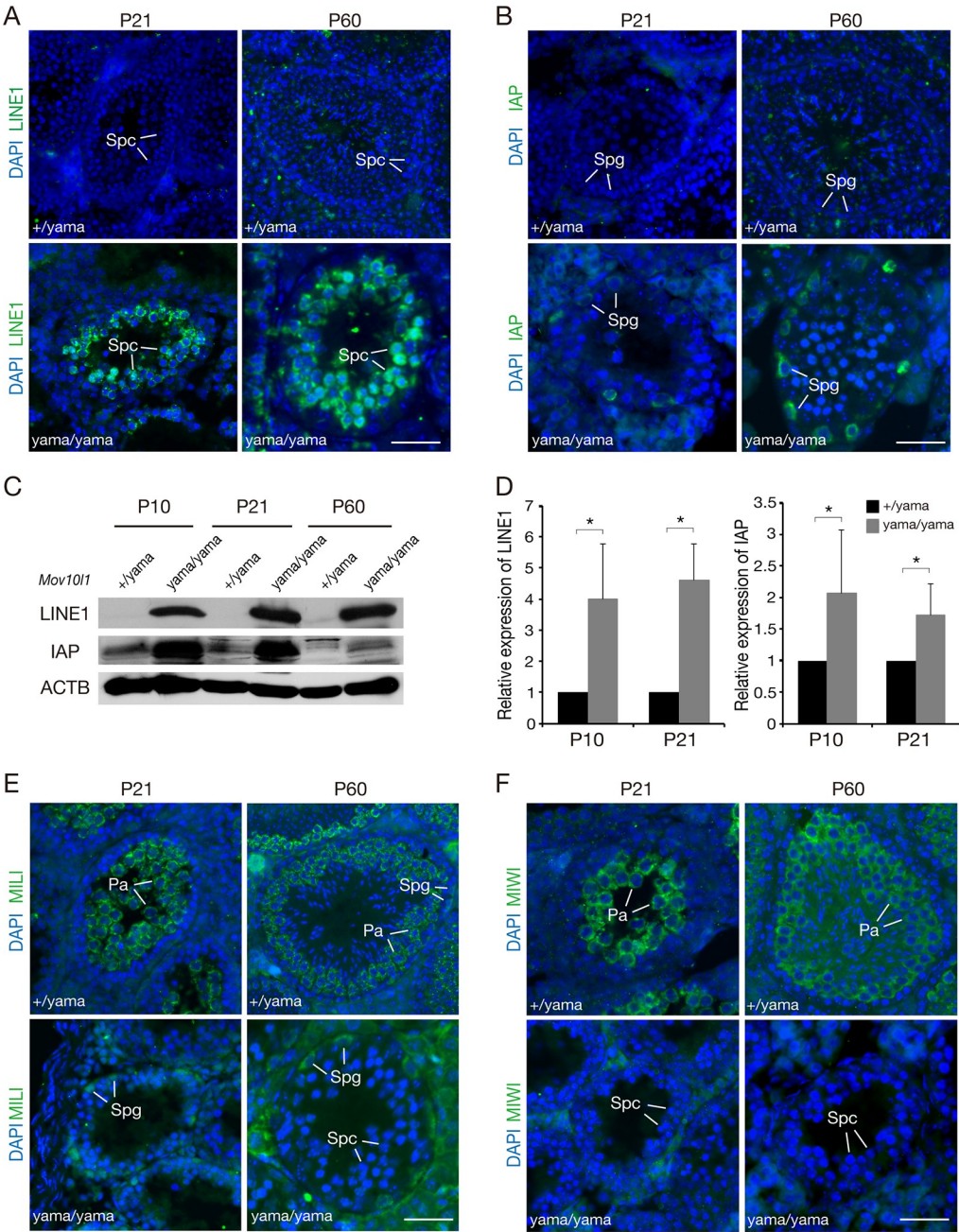

**Fig 3. De-repression of retrotransposons in *Mov10l1*<sup>yama/yama</sup> testis.** Sections of testes from P21 and P60
*Mov10l1*<sup>yama/yama</sup> and control males were immunostained with anti-LINE1 ORF1 (A) and anti-IAP GAG (B) antibodies
[68]. DNA was stained with DAPI. (C) Western blot analysis of LINE1 ORF1 and IAP GAG proteins in testes. ACTB
serves as a loading control. (D) Quantitative RT-PCR analysis of LINE1 and IAP transcripts (mean ± s.d.) in P10 (n = 4)
and P21 (n = 3) testes. *, P < 0.05. (E, F) Sections of testes from P21 and P60 *Mov10l1*<sup>yama/yama</sup> and control males were
immunostained with anti-MILI (E) and anti-MIWI (F). Abbreviations: Spg, spermatogonia; Pa, pachytene
spermatocytes; Spc, spermatocytes. Scale bars, 50 μm.

## *Mov10l1*<sup>fl/yama</sup> *Ngn3*-Cre males display round spermatid arrest

The *Mov10l1*<sup>yama/yama</sup> testes are devoid of normal pachytene spermatocytes and thus are
expected to lack pachytene piRNAs (Fig 2D and 2E). To assess the function of MOV10L1 N-

terminal region in pachytene piRNA production, we bypassed the early meiotic block by using a *Mov10l1*[fl] conditional (floxed helicase domain-encoding region) allele and *Ngn3*-Cre mice, in which Cre expression begins in postnatal day 7 [41,55]. Complete postnatal inactivation of *Mov10l1* (*Mov10l1*[fl/-] *Ngn3*-Cre) leads to post-meiotic arrest at the round spermatid stage [45]. We generated *Mov10l1*[fl/yama] *Ngn3*-Cre mice and compared their phenotype to *Mov10l1*[fl/-] *Ngn3*-Cre mice. The testes from *Mov10l1*[fl/yama] *Ngn3*-Cre males were smaller than controls (Fig 4A and 4B). Histological analysis showed that spermatogenesis from P21 (juvenile)

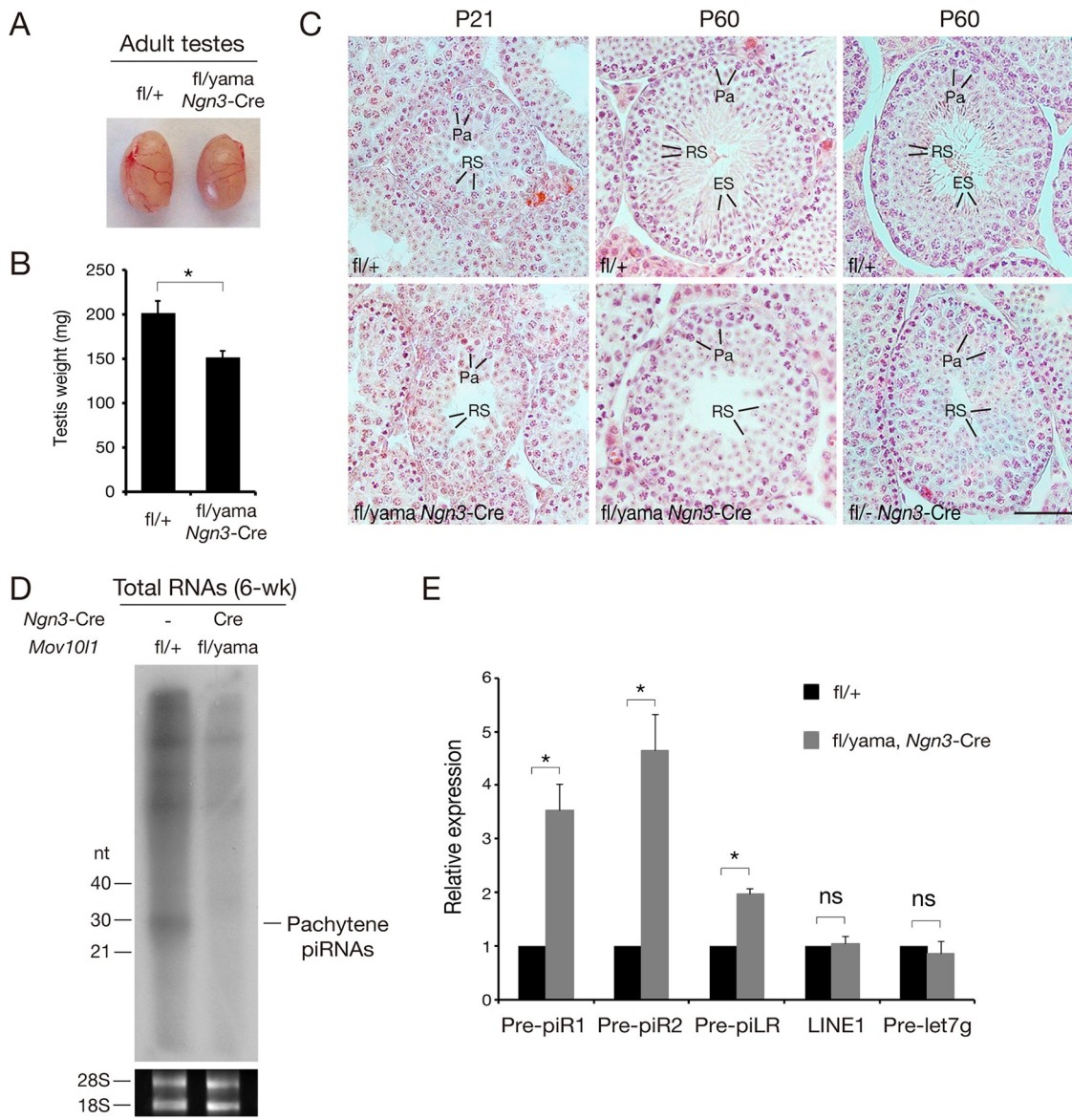

**Fig 4. Round spermatid arrest in adult *Mov10l1*[fl/yama] *Ngn3*-Cre testes.** (A) Images of 8-week-old testes. (B) Reduction of testis weight in 8-week-old *Mov10l1*[fl/yama] *Ngn3*-Cre mice (mean ± s.d; n = 3 per genotype). *, *P* < 0.05. (C) Histological analysis of P21 and P60 *Mov10l1*[fl/yama] *Ngn3*-Cre and P60 *Mov10l1*[fl/-] *Ngn3*-Cre testes. *Mov10l1*[fl/+] testes serve as wild-type controls. Abbreviations: Pa, pachytene spermatocytes; RS, round spermatids; ES, elongated spermatids. Scale bar, 50 μm. (D) Depletion of pachytene piRNAs in 6-week-old *Mov10l1*[fl/yama] *Ngn3*-Cre testes. Total RNAs were [32]P-end-labelled and separated by denaturing polyacrylamide gel electrophoresis. The experiment was done twice with the same result. 28S and 18S ribosomal RNAs serve as loading controls. (E) Quantitative RT-PCR analysis of pachytene piRNA precursor transcripts in 8-week-old testes. Values (mean ± s.d.; triplicates) are the fold change in *Mov10l1*[fl/yama] *Ngn3*-Cre testis normalized to levels in wild-type testis defined as 1. Pre-let7g, a miRNA precursor, serves as a control. *, *P* < 0.005; ns, non-significant (Student's *t*-test).

*Mov10l1*<sup>fl/yama</sup> *Ngn3*-Cre mice progressed to the round spermatid stage as in wild-type (Fig 4C). However, germ cells from P60 (adult) *Mov10l1*<sup>fl/yama</sup> *Ngn3*-Cre mice were arrested at the round spermatid stage, as evidenced by the lack of elongated spermatids in P60 testes (Fig 4C). Therefore, in contrast with the meiotic arrest phenotype observed in *Mov10l1*<sup>yama/yama</sup> testis, *Mov10l1*<sup>fl/yama</sup> *Ngn3*-Cre males displayed post-meiotic round spermatid arrest, similar to that observed in *Mov10l1*<sup>fl/-</sup> *Ngn3*-Cre males (Fig 4C) [45]. The lack of complementation of the targeted *Mov10l1* mutant (*Mov10l1*<sup>-</sup>) allele by the *yama* allele confirms that the phenotype-causing mutation is the detected missense mutation V229E in *Mov10l1*.

## MOV10L1 V229E mutation blocks the primary processing of pachytene piRNA precursors

*Mov10l1*<sup>fl/-</sup> *Ngn3*-Cre testis displays round spermatid arrest and lacks pachytene piRNAs [45]. To assess the production of pachytene piRNAs in *Mov10l1*<sup>fl/yama</sup> *Ngn3*-Cre testes, we isolated and radiolabeled total RNA from testes and found that *Mov10l1*<sup>fl/yama</sup> *Ngn3*-Cre testes were devoid of pachytene piRNAs, which were expected to be ~30 nt (Fig 4D). Pachytene piRNAs are derived from long precursor transcripts from genomic clusters [8,10,56]. Long primary piRNA precursor transcripts are cleaved into intermediate RNAs and processed into mature piRNAs. The absence of pachytene piRNAs in *Mov10l1*<sup>fl/yama</sup> *Ngn3*-Cre testes could be due to the blockage of pachytene piRNA precursor processing, which requires MOV10L1 [45]. To test this possibility, we examined the level of precursors of three pachytene piRNAs: piR1, piR2 and piLR (Fig 4E). qRT-PCR analysis revealed that these three precursors accumulated to 2 to 5-fold higher levels in the mutant testes compared with the wild-type testes (Fig 4E). As expected, the abundance of miRNA precursor Pre-let7g remained constant (Fig 4E). These results demonstrate that the V229 residue of MOV10L1 is essential for the primary processing of piRNA precursor transcripts.

To evaluate whether the absence of pachytene piRNAs leads to de-repression of retrotransposons, such as LINE1 and IAP, in *Mov10l1*<sup>fl/yama</sup> *Ngn3*-Cre germ cells, we immunostained adult wild-type and *Mov10l1*<sup>fl/yama</sup> *Ngn3*-Cre testis sections with anti-LINE1 and anti-IAP antibodies. In comparison with the dramatic increase of retrotransposons in *Mov10l1*<sup>yama/yama</sup> and *Mov10l1*<sup>-/-</sup> testes, LINE1 and IAP were barely detected in *Mov10l1*<sup>fl/yama</sup> *Ngn3*-Cre adult testis (S3A Fig). Therefore, consistent with previous findings in *Mov10l1*<sup>fl/-</sup> *Ngn3*-Cre and other pachytene piRNA pathway mutants [27,45], pachytene piRNAs are not required for repression of LINE1 and IAP retrotransposons.

During the chromatin remodeling process in elongating spermatids, DNA double strand breaks (DSBs) are introduced into the germ cell genome by topoisomerase II beta (TOP2B) to resolve DNA supercoils [57]. The formation of DSBs in elongating spermatids triggers activation of phosphorylation of histone H2AX (γH2AX). Therefore, in wild-type testis, in addition to spermatocytes, γH2AX is present in germ cells that undergo chromatin reconfiguration: in elongating spermatids but not in round spermatids (S3B Fig). However, a dramatic increase of DNA damage visualized by γH2AX was observed in round spermatids from *Mov10l1*<sup>fl/yama</sup> *Ngn3*-Cre testes (S3B Fig). This result was previously observed in *Mov10l1*<sup>fl/-</sup> *Ngn3*-Cre testes [45]. These findings confirm that pachytene piRNAs play a role in maintaining genome integrity in post-meiotic round spermatids.

## Dissociation of MOV10L1 from piRNA pathway proteins in *Mov10l1*<sup>fl/yama</sup> *Ngn3*-Cre testes

We next evaluated the localization of piRNA pathway proteins, including MILI, MIWI and TDRD1 in *Mov10l1*<sup>fl/yama</sup> *Ngn3*-Cre and wild-type testes. In *Mov10l1*<sup>fl/-</sup> *Ngn3*-Cre

spermatocytes, these proteins form a polar conglomeration along with mitochondria [45]. As expected, these three proteins were highly expressed in the cytoplasm of spermatocytes in wild-type testes. Strikingly, these proteins congregated to one pole in the cytoplasm of pachytene spermatocytes in *Mov10l1*<sup>fl/yama</sup> *Ngn3*-Cre testes (Fig 5A–5C), similar to *Mov10l1*<sup>fl/-</sup> *Ngn3*-Cre testes, revealing aberrant localization of the piRNA pathway components. However, MOV10L1<sup>V229E</sup> localized throughout the cytoplasm of spermatocytes and did not display polar aggregation (Fig 5D). In wild-type spermatocytes, piRNA pathway components localize to granules called nuage [58]. As expected, in spermatocytes from control testes, MILI and mitochondria colocalized and were distributed throughout the cytoplasm (S4 Fig). Mitochondria localized exclusively to polar aggregates in spermatocytes from *Mov10l1*<sup>fl/yama</sup> *Ngn3*-Cre testes, showing that along with piRNA pathway proteins, the mitochondria are also abnormally clustered (S4 Fig).

MOV10L1 interacts with MILI, MIWI, and TDRD1 [45]. Since these piRNA pathway proteins were still present in *Mov10l1*<sup>fl/yama</sup> *Ngn3*-Cre testes, we sought to address whether the associations between MOV10L1 and piRNA pathway proteins were affected in *Mov10l1*<sup>fl/yama</sup> *Ngn3*-Cre testes. MOV10L1 was abundant in MILI-immunoprecipitated complexes from wild-type testes but absent in MILI-containing complexes from *Mov10l1*<sup>fl/yama</sup> *Ngn3*-Cre testes (Fig 5E). To confirm this result, we performed reciprocal immunoprecipitations and found that MILI was absent in MOV10L1-immunoprecipitated proteins from *Mov10l1*<sup>fl/yama</sup> *Ngn3*-Cre testes (Fig 5F). Likewise, our co-immunoprecipitation assays revealed that MOV10L1 was sharply decreased in MIWI or TDRD1-immunoprecipitated proteins from *Mov10l1*<sup>fl/yama</sup> *Ngn3*-Cre testes, in comparison with control testes (Fig 5G and 5H).

PLD6 is an endoribonuclease essential for the cleavage of piRNA precursor transcripts [32,33]. Western blot analyses showed that PLD6 was slightly reduced in P14 and P21 *Mov10l1*<sup>fl/yama</sup> *Ngn3*-Cre testes (Fig 5I). To test the association of MOV10L1 and PLD6 in testes, we performed co-immunoprecipitation and found that they formed a complex *in vivo* in wild type (Fig 5J). Consistently, MOV10L1 and PLD6 are found in the same fractions from testis ribosome profiling experiments [59]. However, this association was abolished in *Mov10l1*<sup>fl/yama</sup> *Ngn3*-Cre testes (Fig 5J). These results demonstrate that the association of MOV10L1<sup>V229E</sup> with the piRNA pathway proteins is either abolished or severely reduced, which provides a molecular mechanism underlying the piRNA biogenesis defect.

## The V229E substitution attenuates the MOV10L1-PLD6 interaction

To test whether the V229E mutation directly affects the interaction between MOV10L1 and the piRNA pathway proteins, we co-expressed these proteins in HEK293T cells. Wild-type MOV10L1 interacted with all Piwi proteins (MILI, MIWI, and MIWI2) in transfected cells (Fig 6A–6C). Strikingly, MOV10L1<sup>V229E</sup> was still associated with the Piwi proteins (Fig 6A–6C). MEIOB, a meiosis-specific ssDNA-binding protein required for meiotic recombination, served as a negative control [60,61]. As expected, MOV10L1 did not interact with MEIOB (Fig 6D). These results from co-transfection experiments in cultured cells were in stark contrast with the absent or reduced association of MOV10L1<sup>V229E</sup> with Piwi proteins in testes (Fig 5). It is possible that, in mutant testes, the piRNA biogenesis machinery is disrupted in a way that causes the piRNA pathway proteins to be segregated into different subcellular compartments, thus rendering them unable to interact.

PLD6 plays a conserved role in piRNA biogenesis in *Drosophila* and mice [35,36,62]. Wild-type full-length MOV10L1 co-immunoprecipitated with PLD6 in HEK293T cells, however, the interaction of MOV10L1<sup>V229E</sup> with PLD6 was reduced (Fig 6E). To further investigate the MOV10L1-PLD6 interaction, MOV10L1 was divided into two halves: MOV10L1N (aa 1–708) and MOV10L1C (aa 709–1239) (Fig 6H). Co-transfection experiments in HEK293T cells

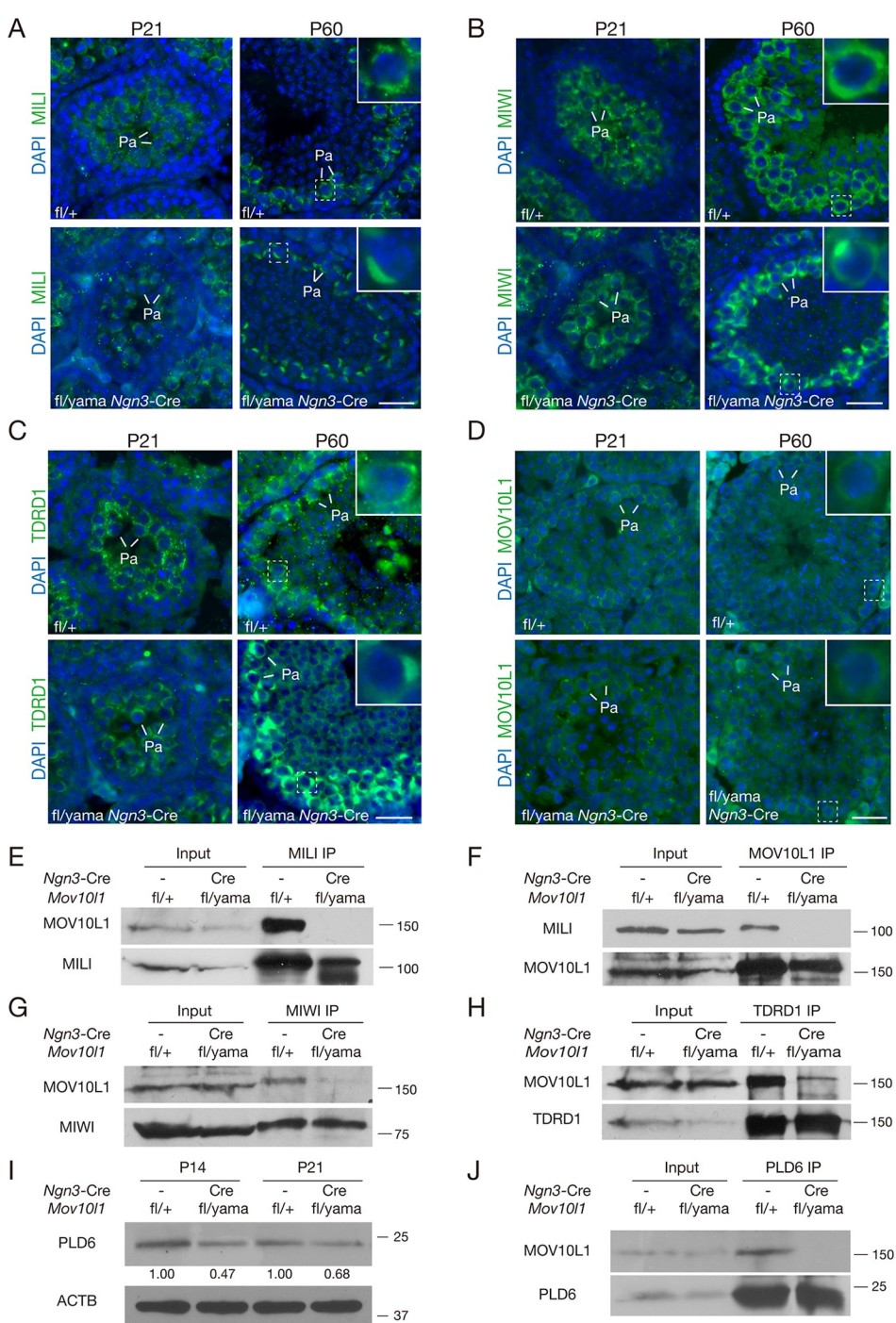

**Fig 5. Dissociation of MOV10L1$^{V229E}$ with MILI, MIWI, TDRD1, and PLD6 in *Mov10l1*$^{fl/yama}$ *Ngn3*-Cre mouse testes.** Polar congregation of piRNA pathway proteins MILI (A), MIWI (B), and TDRD1 (C) in *Mov10l1*$^{fl/yama}$ *Ngn3*-Cre spermatocytes from P21 and P60 testes. (D) Localization of MOV10L1 in *Mov10l1*$^{fl/+}$ and *Mov10l1*$^{fl/yama}$ *Ngn3*-Cre testes. Enlarged view of representative pachytene (Pa) spermatocytes (boxed) are shown in top right corners. Scale bars, 50 μm. (E-J) Co-immunoprecipitation analyses of MOV10L1 with MILI (E and F), MIWI (G), and TDRD1 (H) [69], and PLD6 (J). P21 testes were used for co-IP analyses in panels E, F, H, and J. P28 testes were used for co-IP analyses in G. (I) Western blot analysis of PLD6 in P14 and P21 *Mov10l1*$^{fl/yama}$ *Ngn3*-Cre and control testes.

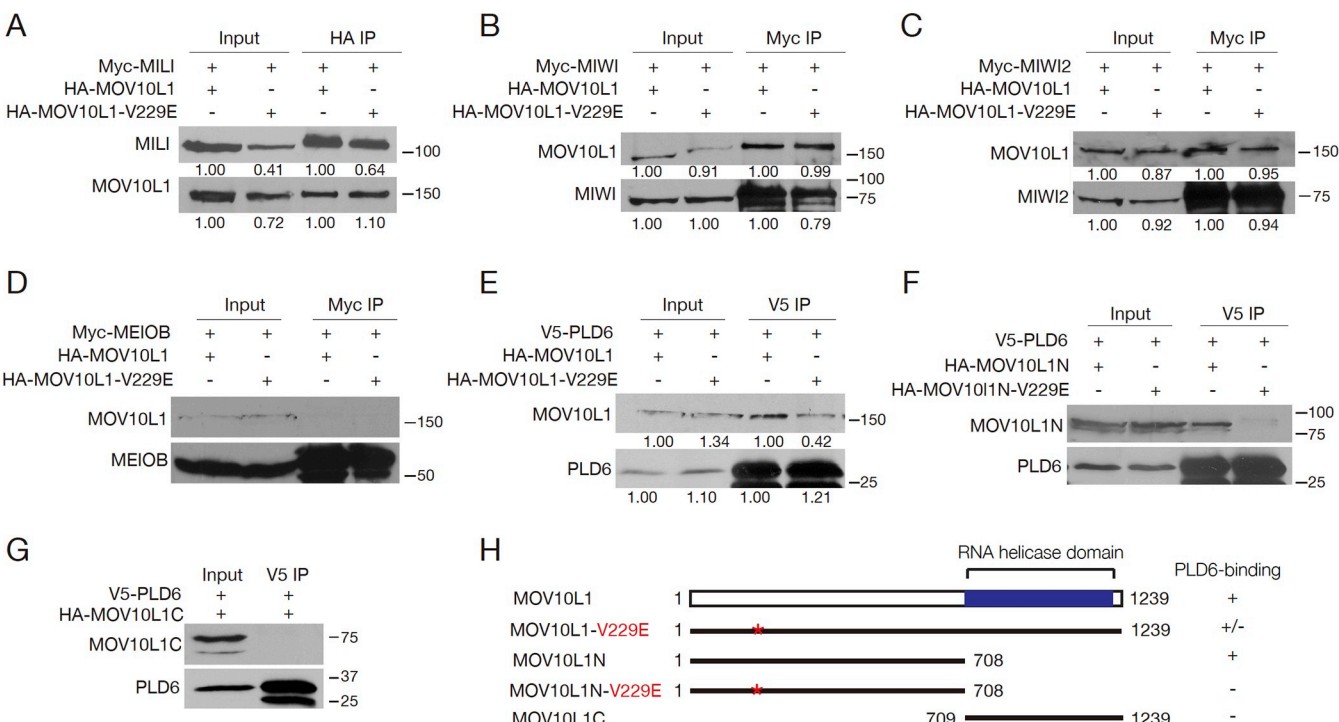

**Fig 6. The V229E substitution reduces the interaction between MOV10L1 and PLD6 in HEK293T cells.** (A-C) Co-immunoprecipitation analyses of MOV10L1 with MILI (A), MIWI (B), and MIWI2 (C). (D) No interaction was observed between MOV10L1 and MEIOB. MEIOB serves as a negative control. (E) Reduced interaction between PLD6 and MOV10L1-V229E. (F, G) Co-immunoprecipitation analyses of PLD6 with MOV10L1N (F), MOV10L1N-V229E (F) and MOV10L1C (G). Band quantification (A, B, C, and E): the band intensity in the 1st and 3rd lanes from left is set at 1.00 in both input and IP; the band intensity in 2nd and 4th lanes is normalized to that in 1st and 3rd lanes respectively. All co-IP experiments were performed twice. Western blotting was performed with anti-HA, anti-MYC, or anti-V5 antibodies. (H) Schematic diagram of the MOV10L1 full-length protein and its variants. The RNA helicase domain is shown. Asterisk denotes the V229E substitution. PLD6-binding: +, strong; +/-, reduced; -, no binding.

showed that PLD6 was associated with MOV10L1N (Fig 6F), but not with MOV10L1C (Fig 6G). However, this interaction was absent between PLD6 and MOV10L1N$^{V229E}$ (Fig 6F). These results imply that the failure in piRNA biogenesis in *Mov10l1* V229E mutant testes could be due to reduced interaction between MOV10L1 and PLD6 (Fig 7).

## Discussion

The MOV10L1 C-terminal RNA helicase activity is essential for piRNA biogenesis (Figs 2A and 7A) [41,42,46]. Here, the identification of the mutation (V229E) in a phenotype-driven ENU mutagenesis screen has allowed us to probe the function of the MOV10L1 N-terminal region. We find that the V229 residue in the MOV10L1 N-terminal region is critical for piRNA biogenesis and spermatogenesis.

Mechanistically, the V229E substitution reduces the interaction between MOV10L1 and PLD6 (Fig 7B). MOV10L1 preferentially binds to RNA G-quadruplex both *in vivo* and *in vitro* [42,43]. Recent biochemical studies have shown that binding to G-quadruplex requires both the N- and C-termini of MOV10L1 [43]. Therefore, we cannot exclude the possibility that the V229E substitution might affect G-quadruplex binding. The MOV10L1 C-terminal region alone does not interact with PLD6. Although MOV10L1N-V229E does not interact with PLD6, the MOV10L1-V229E (full-length) showed reduced interaction with PLD6, suggesting that the MOV10L1 C-terminal region contributes to this interaction but is not sufficient. In our working model (Fig 7B), MOV10L1 binds to piRNA precursors. The V229E substitution inhibits the recruitment of PLD6,

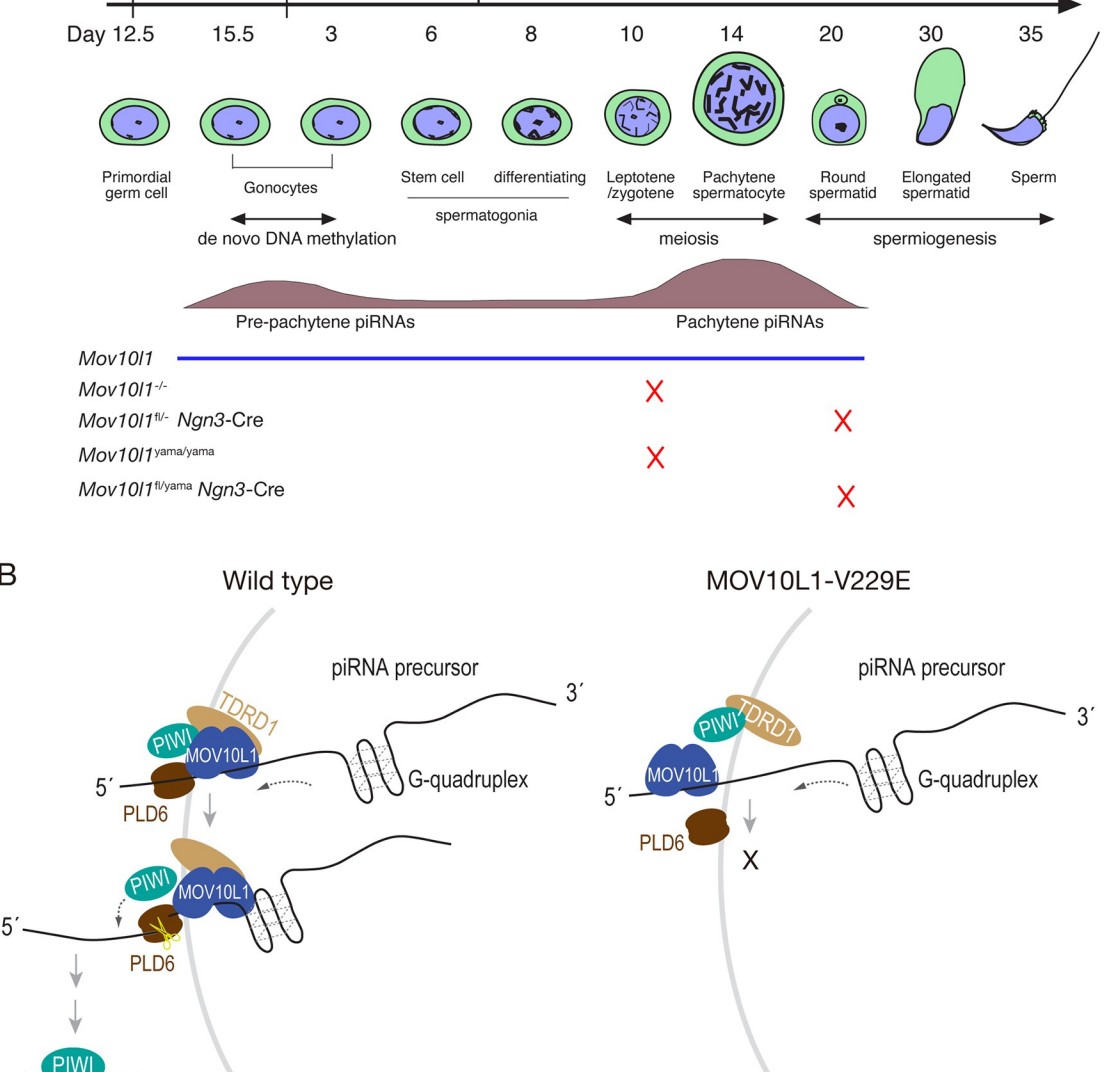

**Fig 7. MOV10L1 V229 residue is critical for processing of piRNA precursors.** (A) A timeline of mouse spermatogenesis. MOV10L1 developmental expression pattern is shown along with two distinct piRNA populations. The point of spermatogenic arrest in each mouse mutant is indicated (X). The *Ngn3*-Cre expression start point is shown. *Mov10l1*[-/-] and *Mov10l1*[fl/-] *Ngn3*-Cre mice with deletion of the RNA helicase domain were reported previously and included for comparison [41,45]. (B) Working model for the essential role of the MOV10L1 V229 residue in the piRNA biogenesis. In wild-type germ cells, MOV10L1 C-terminal region contains the RNA helicase domain and binds to single-stranded piRNA precursors [42]. MOV10L1 interacts with Piwi proteins, TDRD1, and PLD6 to process piRNA precursors. This protein complex translocates on the piRNA precursor in the 5′ to 3′ direction [42,43]. PLD6, a mitochondrial outer membrane-associated endoribonuclease, cleaves the precursor transcript preferentially before RNA secondary structures such as G-quadruplexes to release the 5′ RNA fragment, which is bound by a Piwi protein and processed into a mature piRNA. In this process, the N-terminal half of MOV10L1 recruits PLD6 to the piRNA processing complex by interaction. In *yama* mutant spermatocytes, the V229E mutation disrupts the MOV10L1-PLD6 interaction, leading to a failure in the cleavage of piRNA precursors and lack of association of MOV10L1[V229E] with the mitochondrial membrane.

resulting in a failure of piRNA precursor cleavage and thus a lack of mature piRNAs. As a consequence, piRNA precursor transcripts accumulate in the *Mov10l1* mutant testis.

MOV10L1 interacts with all Piwi proteins [41]. Association of MILI and MIWI with MOV10L1$^{V229E}$ is either absent or dramatically reduced in the *Mov10l1*$^{fl/yama}$ *Ngn3*-Cre testis. However, when co-expressed in HEK293T cells, MOV10L1$^{V229E}$ still interacts with Piwi proteins. What accounts for this difference between *in vivo* (testis) and *in vitro* (HEK293T cells)? Normally, piRNA biogenesis factors localize to cytoplasmic granules called nuage or inter-mitochondrial cement [58]. We postulate that the piRNA biogenesis machinery has collapsed in the *Mov10l1*$^{fl/yama}$ *Ngn3*-Cre testis, due to defects in the very early steps of piRNA biogenesis (Fig 7B), and as a result, the piRNA pathway proteins are redistributed. One possible explanation is that MOV10L1$^{V229E}$ can still interact with Piwi proteins *in vitro* because they are over-expressed. Another possible but non-mutually exclusive explanation is that MOV10L1$^{V229E}$ is physically separated from other proteins such as Piwi and PLD6 in the mutant testes. The latter explanation is supported by the abnormal polar aggregation of MILI, MIWI, and TDRD1 in the cytoplasm of pachytene spermatocytes from the *Mov10l1*$^{fl/yama}$ *Ngn3*-Cre testis. It is further supported by the diffuse cytoplasmic distribution and the lack of polar aggregation of MOV10L1$^{V229E}$ in pachytene spermatocytes from the *Mov10l1*$^{fl/yama}$ *Ngn3*-Cre testis.

Here we report the unusual polar aggregation of piRNA pathway proteins away from their normal location in nuage in *Mov10l1*$^{fl/yama}$ *Ngn3*-Cre testis. This was also observed previously in *Mov10l1*$^{fl/-}$ *Ngn3*-Cre testis [45], *Pld6*$^{-/-}$ testis [35], and *Tdrkh*$^{cKO}$ testis [27]. Although the reason for polar aggregation of piRNA pathway proteins in mutant testes is unknown, it might result from perturbation of piRNA production. Proteins involved in piRNA biogenesis localize to nuage (also called inter-mitochondrial cement or germ cell granule) [58]. Nuages are membraneless electron-dense cytoplasmic condensates. It has been suggested that nuage in germ cells may form via liquid-liquid phase separation [63]. Notably, DDX4, an essential factor for piRNA biogenesis, undergoes phase separation both *in vitro* and in cells [64]. It is possible that disruption of piRNA production causes abnormal phase separation of evenly distributed multiple nuages, leading to fusion into one large polar aggregate per spermatocyte.

The abnormal polar aggregates described previously and those that form in *Mov10l1*$^{fl/yama}$ *Ngn3*-Cre testis colocalized with mitochondria [27,35,45] (S4 Fig). Nuages are located closely to or between mitochondria, and both nuages and mitochondria are important for piRNA biogenesis [58,65]. While the piRNA pathway proteins such as Piwi and MOV10L1 mostly localize to nuage, notably, PLD6, PNLDC1, and TDRKH are mitochondrial outer membrane proteins [26,27,35–37]. TDRKH functions as a scaffold protein to recruit MIWI and PNLDC1 to mitochondria for 3′ trimming of piRNA intermediates [27]. Mitochondria-containing protein extracts are required for reconstitution of piRNA biogenesis *in vitro*, demonstrating that mitochondria play at least a structural role in piRNA biogenesis [65]. The importance of the interaction between MOV10L1, a component of nuage, with PLD6, which resides in mitochondria, for piRNA biogenesis provides another connection between nuage and mitochondria. It is possible that the reduced interaction between MOV10L1V229E and PLD6 causes or contributes to a failure in recruitment of MOV10L1 to mitochondria by PLD6, resulting in perturbation of piRNA biogenesis (Fig 7B).

## Materials and methods

### Ethics statement

All experiments conformed to regulatory standards and were approved by the Institutional Animal Care and Use Committees (IACUC protocol number 806616) of the University of Pennsylvania and Memorial Sloan Kettering Cancer Center (MSKCC).

## Generation of the *Mov10l1*<sup>yama</sup> mutant

The *Mov10l1*<sup>yama</sup> allele was isolated in an N-ethyl-N-nitrosourea (ENU) mutagenesis screen for mutants with autosomal recessive defects in meiosis (Fig 1). Mutagenesis and breeding for screening purposes were conducted at Memorial Sloan Kettering Cancer Center (MSKCC).

The *yama* mutation (T to A) creates a novel *Bse*RI restriction site (S1 Fig). The wild-type and mutant alleles were assayed by primers ACACGACATTGTCAATGCTGTG and GTGGTATGATCTAGTGGAACCAGAA followed by *Bse*RI restriction enzyme digestion at 37˚C for 3 hours. Both alleles produce a 220-bp PCR product and only the mutant PCR product can be digested into 179-bp and 41-bp fragments by *Bse*RI.

The *Mov10l1*<sup>+/-</sup> and *Mov10l1*<sup>fl/fl</sup> mice were previously generated (MMRRC stock number for *Mov10l1*<sup>fl/fl</sup> mice: 036983-UNC) [41]. The *Ngn3*-Cre mice were purchased from the Jackson Laboratory (Stock number: *Neurog3*-Cre, 006333) [55].

## Histological and immunofluorescence analyses

For histological analysis, testes were fixed in Bouin's solution at room temperature overnight, embedded with paraffin and sectioned. Sections were stained with hematoxylin and eosin. In terms of immunofluorescence analysis, testes were fixed in 4% paraformaldehyde (in 1x PBS) for 6 hours at 4˚C, dehydrated in 30% sucrose (in 1x PBS) overnight and sectioned. For surface nuclear spread analysis, testicular tubules were extracted in hypotonic treatment buffer (30 mM Tris, 50 mM Sucrose, 17 mM Trisodium Citrate Dihydrate, 5 mM EDTA, 0.5 mM DTT, 1 mM PMSF). Cells were suspended in 100 mM sucrose and were then spread on a slide which was soaked in paraformaldehyde solution containing Triton X-100.The sections were blocked with 10% goat serum at room temperature for 1 hour and then incubated with primary antibodies at 37˚C for 3 hours. The sections were washed with 1xTBS three times and then incubated with a fluorescein secondary antibody (Vector Laboratories) at 37˚C for 1 hour. After three washes, mounting medium with DAPI (H-1200, Vector Laboratories) was added to the sections. The primary and secondary antibodies used for immunofluorescence analyses were listed in S1 Table.

## RNA extraction and detection of pachytene piRNAs

Total testis RNA was extracted with Trizol (Invitrogen) according to the manufacturer's protocol. 1 µg RNA was dephosphorylated and radiolabeled as described previously [20].

## Reverse transcription and quantitative real-time PCR

1 µg RNA was treated with DNase I and reverse transcribed to cDNA with Superscript II reverse transcriptase (ThermoFisher Scientific). The primers for real-time PCR were listed in S2 Table. Each sample was assayed in triplicates. Quantification was normalized to *Actb* using the ΔCt method.

## Expression constructs

The MOV10L1, MILI, MIWI and MIWI2 expression constructs were previously described [41]. PLD6 expression plasmid was constructed by subcloning mouse PLD6 cDNA to pcDNA3 vector harboring a C-terminal V5 tag. MOV10L1-V229E mutation was generated by overlapping PCR [66] and subcloned into the pCI-neo vector harboring an N-terminal HA tag. MOV10L1 truncated plasmids were engineered by subcloning into pCI-neo vector. All the constructs were verified by Sanger sequencing on an ABI 3730 DNA analyzer.

## Cell culture and transfections

HEK293T cells were maintained in DMEM/high glucose (Mediatech) supplemented with 10% FBS (Sigma) and penicillin/streptomycin (Invitrogen). Plasmid DNA transfections in HEK293T cells were carried out using a standard calcium phosphate method. Briefly, transfections were performed when HEK293T cells were 80% confluent. The transfection mixture (2 μg plasmid, 12.5 μl 2 M $CaCl_2$, 85 μl $ddH_2O$ and 100 μl 2xHEPES, pH 7.15) was incubated at room temperature for 30 min before being added to one well of a 6-well plate in a dropwise manner. Cells were collected 24–36 hours after transfection for further analysis. For cycloheximide treatment experiment, 24 hours after transfection, the cells were treated with cycloheximide (20 μg/ml) for 24 hours to inhibit *de novo* protein synthesis. Cells were collected in 2xSDS PAGE buffer for immunoblotting analysis.

## Co-immunoprecipitation and immunoblotting assays

$1x10^7$ cells were collected after transfection for *in vitro* co-immunoprecipitation and 2 pairs of P21 or P28 juvenile testes were used for *in vivo* co-immunoprecipitation. Cells or testes were lysed in 1 ml RIPA buffer (10 mM Tris, pH 8.0, 140 mM NaCl, 1% Trion X-100, 0.1% sodium deoxycholate, 0.1% SDS, 1 mM EDTA) supplemented with 1mM PMSF. For immunoprecipitation (IP), 1.5% of the lysates were used as inputs. The remaining lysates were pre-cleared with 15 μl protein G Dynabeads (Thermo Fisher Scientific) for two hours, incubated with 1–2 μg primary antibodies at 4˚C for 1 hour, and then incubated with 30 μl protein G Dynabeads overnight. The immunoprecipitated complexes were washed with the RIPA buffer four times and boiled in 30 μl 2× SDS-PAGE loading buffer for 10 min. 20 μl of supernatants were resolved by SDS-PAGE, transferred onto nitrocellulose membranes using iBlot (Invitrogen), and immunoblotted with indicated antibodies. The primary and secondary antibodies used for co-IP and western blot analyses were listed in S1 Table. Band quantification was performed with ImageJ (Version 1.51).

## Statistics

Statistical analysis was performed with Student's *t*-test.

## Supporting information

**S1 Fig. The *Mov10l1*^yama^ allele sequence.** Exon 5 sequence is shown in red (Reference cDNA sequence: NCBI accession number XM_006521556). The mutation is highlighted in green in exon 5 (GTG229GAG ➔ V229E). Bse*RI* site is underlined: GAGGAG(N)$_{10}$. PCR genotyping primers: Forward primer is highlighted in yellow. Reverse primer is highlighted in magenta. (DOCX)

**S2 Fig. Progressive depletion of germ cells in aged *Mov10l1*^yama/yama^ males.** Histology of testes from 4-month and 7-month-old males. Asterisks indicate tubules with Sertoli-cell-only phenotype or severe depletion of germ cells. Scale bar, 50 μm. (TIF)

**S3 Fig. Immunofluorescence analysis of LINE1, IAP and γH2AX in *Mov10l1*^fl/yama^ *Ngn3*-Cre males.** (A) Sections of testes from 6-week-old wild-type and *Mov10l1*^fl/yama^ *Ngn3*-Cre males were immunostained with anti-LINE1 and anti-IAP antibodies. *Mov10l1*^-/-^ (knockout) testis serves as a positive control [41]. (B) Presence of γH2AX in round spermatids from 6-week-old *Mov10l1*^fl/yama^ *Ngn3*-Cre testes (right panels). Elongating spermatids in *Mov10l1*^fl/+^ (control) stage XI tubules are γH2AX-positive (left panels). Round spermatids in *Mov10l1*^fl/+^

(control) early stage (before IX) tubules are γH2AX-negative (middle panels) but round spermatids in *Mov10l1*<sup>fl/yama</sup> *Ngn3*-Cre testes (right panels) are γH2AX-positive. Abbreviations: Spg, spermatogonia; Spc, spermatocytes; RS, round spermatids; ES, elongating spermatids; Zyg, zygotene spermatocytes; Pa, pachytene spermatocytes; Dip, diplotene spermatocytes. Scale bars, 25 μm.
(TIF)

**S4 Fig. Colocalization of mitochondria with polar aggregates in pachytene spermatocytes from *Mov10l1*<sup>fl/yama</sup> *Ngn3*-Cre testes.** Testis sections from P60 *Mov10l1*<sup>fl/+</sup> (control) and *Mov10l1*<sup>fl/yama</sup> *Ngn3*-Cre mice were immunostained with anti-MILI antibody and OXPHOS. OXPHOS is a cocktail of five monoclonal antibodies against mitochondrial proteins (S2 Table). MILI and mitochondria colocalize but are distributed throughout the cytoplasm in *Mov10l1*<sup>fl/+</sup> pachytene spermatocytes. However, only polar aggregates in *Mov10l1*<sup>fl/yama</sup> *Ngn3*-Cre pachytene spermatocytes are OXOPHOS-positive. Representative polar aggregates are indicated by arrows. Inset shows an enlarged view of the boxed spermatocyte. Scale bar, 25 μm.
(TIF)

**S1 Table. Primary and secondary antibodies.**
(DOCX)

**S2 Table. Real-time PCR primers.**
(DOCX)

## Acknowledgments

We thank Ramesh S. Pillai for LINE1 ORF1 antibody, Bryan Cullen for anti-IAP antibody, Shinichiro Chuma for TDRD1 antibody, Prabhakara Reddi for anti-SP10 (ACRV1) antibody, Joseph Baur and Narayan Avadhani for mitochondrial protein antibodies. We thank Wang lab members (Jessica Chotiner, Rui Guo, Rong Liu, and Fang Yang) for critical reading of the manuscript. We thank Keeney lab members Luis Torres and Jacquelyn Song for assistance with genotyping and mouse husbandry. We thank the Genetic Analysis Facility (Centre for Applied Genomics, Hospital for Sick Children, Toronto, ON, Canada) for microarray analysis. For whole-exome sequencing, we thank Nathalie Lailler at the MSKCC Integrated Genomics Operation.

## Author Contributions

**Conceptualization:** Yongjuan Guan, Scott Keeney, Devanshi Jain, P. Jeremy Wang.

**Data curation:** Yongjuan Guan, Devanshi Jain.

**Formal analysis:** Yongjuan Guan, Scott Keeney, Devanshi Jain, P. Jeremy Wang.

**Funding acquisition:** Scott Keeney, Devanshi Jain, P. Jeremy Wang.

**Investigation:** Yongjuan Guan, Devanshi Jain.

**Methodology:** Yongjuan Guan, Devanshi Jain.

**Supervision:** Scott Keeney, P. Jeremy Wang.

**Validation:** Yongjuan Guan, Scott Keeney, Devanshi Jain, P. Jeremy Wang.

**Writing – original draft:** Yongjuan Guan, Devanshi Jain, P. Jeremy Wang.

**Writing – review & editing:** Yongjuan Guan, Scott Keeney, Devanshi Jain, P. Jeremy Wang.

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
