## [Decision Letter · Decision Letter 0]

22 Dec 2020

Dear Jeremy,

Thank you very much for submitting your Research Article entitled 'yama , a mutant allele of Mov10l1, disrupts retrotransposon silencing and piRNA biogenesis' to PLOS Genetics.

The manuscript was fully evaluated at the editorial level and by independent peer reviewers. The reviewers were unanimous in their praise for this work, and I concur fully that this is a beautiful manuscript. However, each reviewer  identified some very minor concerns that we ask you address in a revised manuscript. We therefore ask you to modify the manuscript according to the review recommendations. Your revisions should address the specific points made by each reviewer.

[LINK]

Yours sincerely,

Paula E. Cohen

Associate Editor

PLOS Genetics

Gregory Barsh

Editor-in-Chief

PLOS Genetics

Reviewer's Responses to Questions

**Comments to the Authors:**

Reviewer #1: Thank you for submitting this manuscript. The experiments described within aim to characterize the poorly understood function of the N-Terminal domain of MOV10L1, a germ cell specific RNA helicase essential for piRNA biogenesis. The authors describe a single amino acid substitution in the N-Terminal region they call “yama”, that causes meiotic arrest and male sterility. To dissect Mov10L’s role in pre-pachytene and pachytene piRNAs, the authors cleverly designed a conditional mouse model shifting the mutation to the later piRNA biogenesis stage, also resulting in meiotic arrest but with distinct difference. This work provides more mechanistic understanding of both pre-pachytene and pachytene biogenesis and provides a functional link between MOV10L and PLD6.

MAJOR:

Fig2D: How is the SYCP1/3 defect linked to the MOV10L deficiency? Is this simply a readout? If the defect is interpreted as a block or delay in meiosis, then a timecourse showing the previous stages would be expected to be identical at zygotene, just a stall. Is this correct? Otherwise, the term “zygotene-like” is appropriate but then also needs to be supported. If HET and KO zygotene are identical, then the interpretation needs to be adjusted slightly. If this is a blockade, would you expect an increase in Zygotene presenting cells? Or is the implication that these zygotene-like cells are directly shuttled into an apoptotic pathway? A quantification of zygotene accumulation, linked to the increased TUNEL staining in 2F would be a decent link/bridge for this interpretation. Perhaps a simple ATM or ATR staining on spreads would support the checkpoint suggestion in line 140.

Fig2G and H: The stability hypothesis was addressed nicely, but can you elaborate more on why such a noticeable difference in abundance is seen between p21 and p60? Is the “first wave” intrinsically distinct in this particular analysis? Is there a different proportion of spermatocytes between these stages? I don’t think so but please correct me. Presumably pachytene is depleted in both ages similarly. Could another explanation be that the defects become more pronounced over time? Would a 90 or 120 day mouse have even less? I would consider exploring this briefly to link this phenotype to ae related loss of fertility in humans. Could you also confirm the antibody was not raised against this particular region (silly control but worth asking)? Otherwise I would imagine there are in vivo post translational modification and regulation explaining this.

Fig3A and B: Does the localization of upregulated LINE1 (spermatocytes) and IAP (Spermatogonia) correspond properly with what is expected from Mov10L(yama) defect? The Zheng 2010 references suggests similar pattern, but can you comment on the severity? Is this a phenocopy? If so, how much of the null mutation can you attribute to the N-terminal mutation and previous studies describing the helicase domain?

Fig4: The previous set of experiments used HET vs KO. The following experiments use WT vs fl/yama,Cre+. Is the reason for this the fl/yama,Cre- animal is compromised in some way? I might just be confused. I think a Supp figure diagram would really help me understand.

Figure 6: I wonder if the persistent binding is simply technical? Have you tried using the MYC or HA antibodies for exposing the protein? Maybe an un-tagged prey? In Vitro CO-IP are something voodoo. It is also the wrong species and the wrong cell type. Are you in possession of a reliable mouse cell line that can be transfected? I can only think of ES Cells or pMEFs at the moment. The in vivo testis data is more important to me in any case.

Model: I liked the connection between the phenotype and the aggregates seen in figure 5, but I feel like there wasn’t enough exploration into this observation to generate the model. It is left mostly speculative at this point. The model shows mitochondrial involvement but no experiments were done to make that link. The study and model suggests an accumulation of piRNA precursor transcripts and the aggregates suggest this is happening at or around the mitochondria. The mitochondria normally exist all throughout cells (unless germ cells are different) so you could imagine a mitochondrial aggregate as well. I think to strengthen this link either RNA-FISH (piRNA precursor), or a co-localization study of the aggregates and a mitochondrial marker would convincing. It would also provide a possible link to metabolic disfunction, since a clumped up mitochondria is an unhappy mitochondria. Work by David Chan at Caltech might be worth a look.

MINOR:

-In general, can you provide explanation for experiments where one of the p10, p21 and p60 timepoints are not presented? There seems to be a bit of jumping around in their inclusion throughout the manuscript.

-Line 51: How conserved are mouse and human across the entire length of MOV10L (overall % conservation)?

-Line 60: Typo, should it be PIWIL4?

-Line 96: Very cool!

-Line 106: Present pdf makes it very hard to see details in A.

-Line 107: Can you elaborate at least one sentence on “meiotic defects”? What is solely based on SYCP3/yH2AX?

-Line 116: Danio Rerio has an L in position 229. Would a leucine in this position be predicted to be functionally equivalent? Can this be shaded differently, be given a conservation score or simply remove zebrafish? Also, dos this region share ANY similarity to a known domain or is all this space seemingly room to interact with the various PIWI pathway components?

-Line123: Can you label the type of histological staining on Fig1D? Legends just say “Histology”. Presumably H&E.

-Line160: Please avoid the use of “significant” unless associated with statistics. While obvious from the representative images, there may be regions of the section with similar mis-regulation due to age or other unrelated defects.

-Line 164: What is used as a baseline in Fig 1D? The Western blot appears blank in WT. Is there a reason why p60 is excluded? Can you add statistics to this observation (het is missing error bars). Also, the IAP background in the WT control lane appears higher than expected. Is this consistent with the literature? Is this possibly a consequence of the mixed background used (or has this been sufficiently backcrossed to make that not a concern?)

-Line 165: The statement isn’t completely supported. You cannot say they are not silenced without a comparison to a system where retrotransposon regulation is completely shut down. You can revise this to say something like “de-repressed”.

-Line 172: For 3F, The staining of MIWI in p60 is a bit more unusual than described. Is this representative? The surrounding tissues seems fairly positive for signal (I see a bit of this in 3E as well). Also, there is low level staining and two spots that might be nuclear (are they)? If this is not representative, please consider a different image. Do you have a lower magnification?

-Line 172: Can you add a sentence as to the significance of this finding? If it is totally expected, can you say how much of your expectations were matched? Should this totally phenocopy the null mutant?

-Line 194: I think the labeling throughout figure 4 is a little inconsistent. Sometimes NGN3 is shown. Something is it just “Cre”. Also, the control in 4D is a fl/+ mouse. Is this a typo or are there different controls throughout this figure?

-Line 199: Please provide error bars and statistics (student’s t-test. Show SEM assuming this was done in biological replicates). The Pre-piLR significance is unclear.

-Line205: Is the lack of LINE1 expression accurately controlled? I would consider evaluating whether the specific LINE1 family primers actually corresponds to an intact and functional ORF1p generating LINE1 subfamily. You have the perfect control actually. Can you show an increase in LINE and IAP through qPCR on the testes showing in Figure 3A and 3B (this would be nice to include in figure 3 anyway).

Line 217: S2b legends are written in a slightly confusing way. Please consider revising and confirm they properly correspond to image.

-Line 227: Is there any appreciable difference in MOV10L1 localization of abundance in any of the mutants from this analysis?

-Line 244: I think it should be (Fig. 5J) to show abolished association.

-Line 261: Very minor, but please consider moving the image of 6h earlier in the image so that the figure citations are in order. (Turn 6H into 6F)

-Line 628: hard to understand, please consider revising. Could “phenotypically WT” technically be mutated at this region?

Reviewer #2: This manuscript by Guan et al., identifies a single amino acid point mutation in the piRNA biogenesis factor called MOV10L1. Previous studies have shown that MOV10L1 is RNA helicase, and this RNA helicase activity residing in the C-terminal part of the protein is absolutely essential for piRNA biogenesis, and as a consequence is required for transposon silencing in the mouse male germline, and is required for male fertility. In this study, they identify a point mutation in the highly conserved N-terminal region, whose functions are not known. The authors demonstrate that the mutant protein is expressed, albeit at reduced levels, but is unable to support the normal functions of the protein. Using some clever genetics approaches, they examine the impact of the mutation in two different developmental windows during spermatogenesis, when two distinct populations of piRNAs are expressed.

This is an elegant study that combines genetics and biochemistry, and for the first time sheds light on the non-catalytic role of MOV10L1 in piRNA biogenesis. It uses its N-terminal domain to assemble the piRNA biogenesis machinery, by directly recruiting the endonuclease responsible for 5’ end generation of piRNAs. This finding will have huge implications for our current model of how the complex is assembled. I support its publication.

Minor comments:

Figure 5D: Why is the yama version of Mov10L1 not accumulating in the unique crescent-shaped structures? One would have expected Mov10L1 to be in these structures like MILI, MIWI and TDRD1. Is it because of an antibody issue? A mention of this in the discussion would be good.

Figure 5D: It is interesting that yama mutant Mov10L1 fails to interact with PLD6. It would be interesting to localize PLD6 and see if it is also in the crescent structures. I would imagine that it should not be there, as it fails to interact with Mov10L1. I am not sure if suitable antibodies are available, but if this data can be obtained, it could be added.

Reviewer #3: In this manuscript the authors report on a new mutation in the Mov10l1 gene, affecting structure of the N-terminal region, for which the function was previously unknown. They find that the mutation causes male infertility, derepression of LINE1 and IAP retrotransposons, and aberrant processing of both pre-pachytene and pachytene piRNAs. An apparent primary cause of the aberrant piRNAs is failure of the mutated protein to bind to PLD6, a key endoribonuclease in the piRNA processing pathway, thereby assigning functional significance to the N-terminal domain of the protein. As a summary evaluation, the manuscript is clearly written, the figures are excellent (gorgeous histology and IF!), the conclusions are supported by the results, and the overall findings are of significant interest.

This study was based on the phenotype of a recent ENU-induced mutation, “yama,” in the Mov10l1 gene. The gold standard for ENU mutagenesis programs is to prove that the detected mutation is causative of the phenotype (rather than an undetected or under-appreciated regulatory SNP in nearby sequence). The authors provide robust evidence that the yama mutation is Mov10l1 is the sole cause of the phenotype. First, the parameters of homozygous yama infertility phenocopy the original Mov10L1 knockout mutation; second and most definitively, the yama infertility phenotype does not complement a conditional Mov10l1 allele (hets are infertile, revealing MOV10L1 function post-pachynema); and third, biochemical analyses reveal the mutant yama MOV10L1 protein lacks protein interactions (specifically with PLD6) typical of wildtype MOV10L1.

Overall, the manuscript is excellent: the quality of the figures is high and the data strongly support the conclusion of aberrant function of the yama mutant MOV10L1. There are, however, some points that the authors should consider in order to improve clarity:

Fig. 1A Legend (p. 20, line 616): It is stated that 1/8 of the G3 males are expected to be hom, but this should be clarified to state that 1/8 of all individuals are both male and hom (or that ¼ of males are hom).

Fig. 3B: What are the intensely staining cells in the lower left? These appear to be interstitial cells (???).

Fig. 4: It would be nice, though not necessary, to have images of the conditional hom Mov10l1 flox-Ngn3-Cre.

p. 7, line 187: The word “lied” should be changed to “is the detected missense mutation V229E.”

p. 8, lines 214-216: This sentence needs to be clarified; it doesn’t make sense as written, especially since gH2AX typically localizes to spermatocytes.

p. 8, line 226: The localization to one pole does not in itself indicate that the piRNA path is disturbed but does reveal aberrant localization of the components.

Fig. 7B and associated text about the proposed mode of action of MOV10L1 and PLD6: Fig. 7B is helpful in portraying the proposed pathway and yama defect with the exception of lack of clarity with respect to the line labeled “mitochondrion.” This line (and label) is not described in the legend, and although localization of granules and nuage to mitochondrial cement is mentioned in the text, the role of mitochondria and/or the mitochondrial membrane (is that what the gray line is?) per se is not discussed. Indeed, the Discussion would be more impactful with a bit more attention to the still-evolving concepts of nuage, the aberrant polar accumulation of piRNA pathway proteins (but not MOV10L1) in conditional mutants, and the significance of localization to subcellular domains of mitochondrial cement.

**Have all data underlying the figures and results presented in the manuscript been provided?**

Reviewer #1: Yes

Reviewer #2: Yes

Reviewer #3: Yes

PLOS authors have the option to publish the peer review history of their article (what does this mean?). If published, this will include your full peer review and any attached files.

Reviewer #1: No

Reviewer #2: No

Reviewer #3: No

---

## [Editor Report · Decision Letter 1]

9 Feb 2021

Dear Jeremy and Devanshi,

We are pleased to inform you that your manuscript entitled "yama, a mutant allele of Mov10l1, disrupts retrotransposon silencing and piRNA biogenesis" has been editorially accepted for publication in PLOS Genetics. Congratulations!

Yours sincerely,

Paula E. Cohen

Associate Editor

PLOS Genetics

Gregory Barsh

Editor-in-Chief

PLOS Genetics

Comments from the reviewers (if applicable):

**Data Deposition**

http://datadryad.org/submit?journalID=pgenetics&manu=PGENETICS-D-20-01770R1

**Press Queries**

---

## [Editor Report · Acceptance letter]

22 Feb 2021

PGENETICS-D-20-01770R1 

*yama*, a mutant allele of *Mov10l1*, disrupts retrotransposon silencing and piRNA biogenesis 

Dear Dr Wang, 

We are pleased to inform you that your manuscript entitled "*yama*, a mutant allele of *Mov10l1*, disrupts retrotransposon silencing and piRNA biogenesis" has been formally accepted for publication in PLOS Genetics! Your manuscript is now with our production department and you will be notified of the publication date in due course.

With kind regards,

Alice Ellingham

PLOS Genetics

On behalf of:
